# Impact of The Daily Mile on children's physical and mental health, and educational attainment in primary schools: iMprOVE cohort study protocol

Bina Ram ,[1] Anna Chalkley ,[2] Esther van Sluijs,[3] Rachel Phillips ,[1] Tishya Venkatraman ,[1] Dougal S Hargreaves,[1,4] Russell M Viner ,[5] Sonia Saxena [1]

¹Faculty of Medicine, School of Public Health, Imperial College London, London, UK
²School of Sport, Exercise and Health Sciences, Loughborough University, Loughborough, UK
³MRC Epidemiology Unit, Cambridge University, Cambridge, UK
⁴Department for Education, London, UK
⁵Institute of Child Health, University College London, London, UK

**Correspondence to**
Dr Bina Ram;
b.ram@imperial.ac.uk

## ABSTRACT

**Introduction** School-based active mile initiatives such as The Daily Mile (TDM) are widely promoted to address shortfalls in meeting physical activity recommendations. The iMprOVE Study aims to examine the impact of TDM on children's physical and mental health and educational attainment throughout primary school.

**Methods and analysis** iMprOVE is a longitudinal quasi-experimental cohort study. We will send a survey to all state-funded primary schools in Greater London to identify participation in TDM. The survey responses will be used for non-random allocation to either the intervention group (Daily Mile schools) or to the control group (non-Daily Mile schools). We aim to recruit 3533 year 1 children (aged 5–6 years) from 77 primary schools and follow them up annually until the end of their primary school years. Data collection taking place at baseline (children in school year 1) and each primary school year thereafter includes device-based measures of moderate-to-vigorous physical activity (MVPA) and questionnaires to measure mental health (Strengths and Difficulties Questionnaire) and educational attainment (ratings from 'below expected' to 'above expected levels'). The primary outcome is the mean change in MVPA minutes from baseline to year 6 during the school day among the intervention group compared with controls. We will use multilevel linear regression models adjusting for sociodemographic data and participation in TDM. The study is powered to detect a 10% (5.5 min) difference between the intervention and control group which would be considered clinically significant.

**Ethics and dissemination** Ethics has been approved from Imperial College Research Ethics Committee, reference 20IC6127. Key findings will be disseminated to the public through research networks, social, print and media broadcasts, community engagement opportunities and schools. We will work with policy-makers for direct application and impact of our findings.

## INTRODUCTION

Increasing physical activity during childhood is a global goal to improve health in the population and reduce the growing health risks from non-communicable diseases.[1 2] Regular physical activity is associated with better cardiorespiratory and muscular fitness and bone health.[3] A growing science base suggests links between children's physical activity and psychological and social health including sleep, cognition, self-confidence and social interaction.[3] This is particularly important in childhood where health behaviours may become embedded.[4] The UK's chief medical officers recommend that children and young people, aged 5–18 years, should engage in sport and physical activity for an average of 60 min or more every day.[5] The latest national self-report activity figures for children and young people by Sport England showed that around 47% are meeting these targets in the UK while 29% participate in less than an average of 30 min a day.[6]

School-based active mile initiatives are widely publicised to address significant shortfalls in children's physical activity.[7] Schools are considered ideal settings for physical activity initiatives given their wide reach.[8 9] The Daily Mile (TDM) is a school-based active mile initiative that was launched in 2012 by a head teacher in Scotland, UK, who wanted to increase the fitness of her pupils.[10] Since 2012, one in five schools in England are now registered with TDM[11] and it has a growing

global adoption.[12] TDM is a teacher-led, free and simple initiative that involves children running or jogging for 15 min ('a mile') at least three times a week, in addition to national curriculum physical education (PE) and scheduled break times. TDM has been promoted in the UK's Childhood Obesity Plan as an easy and accessible way to increase children's physical activity during the school day to improve their fitness, health and well-being.[13] As schooling is compulsory for all children in England, school-based active mile initiatives have the potential to reach all children, including those living in poverty, which may help to reduce the gap in physical activity among children.[14]

## Strengths and limitations of this study

► This quasi-experimental study is the most comprehensive study to date to examine the longitudinal impact of The Daily Mile on children's health.
► Including schools in Greater London will reflect the diverse multiethnic primary school population.
► Low response rates from school surveys may result in a biased sample.
► Attrition may affect precision of effect sizes and bias results.

TDM appears to be reaching a nationally representative primary school population. Up to August 2019, 22% of all state-funded primary schools in England were registered, reaching a population of just over 963 000 children.[11] The evolving evidence base of the benefits of TDM on physical activity is promising; a study by Chesham et al[15] showed an increase in children's moderate-to-vigorous physical activity (MVPA) by approximately 9 min,[15] and a study by Morris et al[16] demonstrated that during one active mile lesson, children accumulated between 5 and 15 min of MVPA. While there are acute assessments suggesting that children accumulate around an average of 10 min of MVPA during TDM, there is limited evidence of its impact on children's mental health[17 18] and inconsistencies on benefits to academic performance.[16 19] Furthermore, some of the previous studies have focused on weight-related outcomes such as body fat and body mass index (BMI) and have reported conflicting results.[15 20] TDM was not designed as an obesity prevention initiative and there is inconclusive evidence that increased physical activity alone is effective in preventing obesity in children.[21] The evidence base still lacks an assessment of the impact and sustainability of TDM throughout primary school life.

A major gap in the evidence is the lack of data; there is limited evidence on how TDM is implemented in schools outside of experimental studies and limited follow-up times.[22–24] Currently, there are no evaluations of the impact and sustainability of TDM throughout primary school life.

The iMprOVE Study will provide a robust real-world evaluation of the impact of TDM on children's health, well-being and educational attainment over the course of primary school.

## Aims and research questions

The aim of the iMprOVE Study is to compare the change in MVPA of children in TDM schools with non-TDM schools across primary school years.

The following research questions will be addressed:

► What is the difference in MVPA accumulated during the school day between children in TDM and non-TDM schools across primary school years?
► What are the differences in mental health and educational attainment of children in TDM and non-TDM schools across primary school years?

► How much MVPA is attributed to doing TDM and how much does this contribute to children's overall physical activity?
► To what extent do primary schools in Greater London implement TDM according to its core principles?
► Are there any differential impacts of TDM on health in subgroups of children, schools and areas with greater health need?

## METHODS AND ANALYSIS

### Study design

The iMprOVE Study is a longitudinal quasi-experimental cohort study of state-funded primary schools located in Greater London which has a diverse multi-ethnic urban population. The target population is children in schools that participate in TDM (intervention group) compared with those that do not participate in TDM or any other active mile initiative (control group). We will assess children in the first year of primary school (aged 5–6 years) and follow them up annually in each successive primary school year until the end of primary school (year 6, children aged 10–11 years). In total, we will collect six waves of data (figure 1).

### The intervention

TDM involves every child running or jogging at their own pace for 15 min, at least three times a week during the school day. TDM is in addition to curricular lessons including PE and scheduled break times. There are 10 core principles for successful delivery of TDM which schools are encouraged to follow (online supplemental table 1). These include that TDM should be conducted in all weathers and the full 15 min are spent on the activity without having to change clothing or shoes.[25] TDM is also flexible to local adaptation by schools.

Since 2017, we have held several professional and public multi-stakeholder consultations about which of the 10 core principles of TDM were essential criteria for successful implementation. The consultations included healthcare professionals, researchers and members from TDM Foundation. A consensus was reached agreeing a definition for school participation in TDM which was that three essential principles were adhered to: (1) the whole school takes part; (2) children are running or jogging for 15 min and (3) it is implemented at least three times a week.

### Procedure

#### School survey and recruitment

To identify which schools to assign to the intervention and control groups at baseline, we will send a school survey by post (including a link to the online version) with a free-post return envelope to all state-funded primary schools in Greater London (n=1721) derived from our database of primary schools in England.[11] Any teacher who is best placed to answer questions about their school's involvement with TDM can complete the survey. Schools that do not respond will be contacted by email 2 weeks

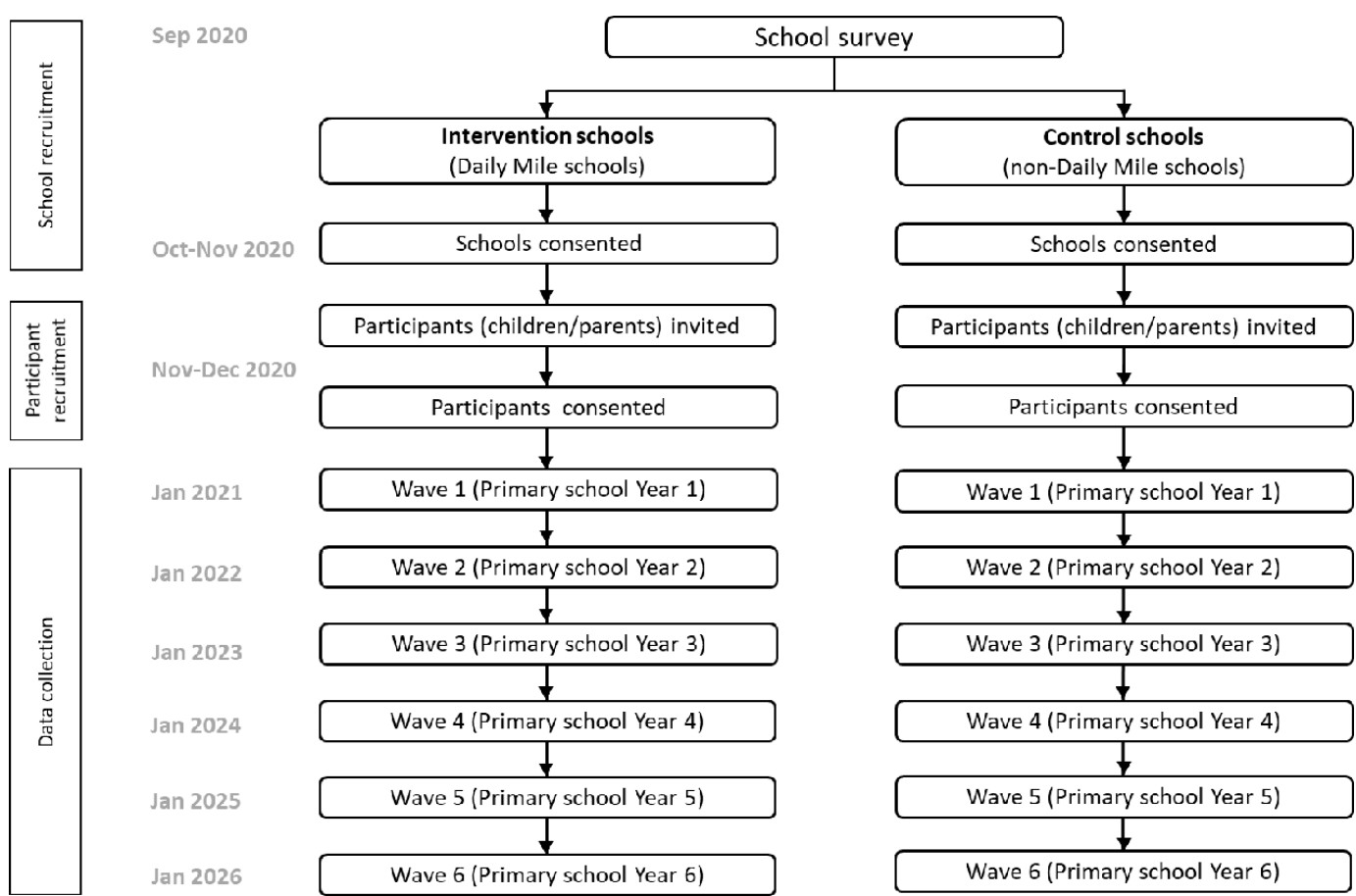

**Figure 1** iMprOVE Study design.

after initial contact with a further email 1 week later. If required, we will conduct one further follow-up by phone to encourage survey completion 2 weeks after the second email. Responses from the survey will identify whether schools participate in TDM and if they do, how many of the 10 principles are adhered to. Schools which participate in the three essential principles of TDM identified from the public and expert consultation will be assigned to the intervention group and schools not participating in TDM or any other active mile initiative will be assigned to the control group. Primary schools where TDM is not implemented by the whole school will be considered ineligible for the study, for example, if they report only some classes or year groups take part, thus meeting only two of the three criteria considered essential (figure 2). We will contact the eligible schools to participate in the next stage of the study where through the schools, we will recruit year 1 primary school children (aged 5–6 years) and their parent/carer by letter and information sheet. For a child to be eligible to take part, their parent/carer will also need to agree to participate.

### Informed consent
Written and informed consent will be sought at the point of recruitment, prior to any data collection. We will first obtain consent from the headteacher of the school to allow the distribution of invitation letters to parents. Parents

who agree to participate in the study will be emailed a consent form (for their participation), a parental consent form (allowing their child to participate) and a child assent form. Consent to take part in the study at baseline (Wave 1) will assume contact for subsequent follow-ups. At each follow-up, we will provide an option for participants to decline further participation.

### Outcomes and measures
We will collect data from children, parents and teachers which are detailed in online supplemental table 2.

The main outcome is children's mean MVPA minutes during the school day (we will match individual children to the start and end times of their school as reported by the teacher) measured using the GENEActiv (Activinsights, Cambridge, UK) wrist-worn accelerometers. The GENE-Activ accelerometer is worn 24/7 including when in water and during sleep. Children will be instructed to wear the monitors for 24 hours on each day of a consecutive 7-day wear period. The accelerometers will be initialised for 9 days to obtain 7 full days of wear (excluding the date of issue and day of collecting the accelerometers). We will use raw acceleration data to classify wear time and assess time spent in activity intensities (including moderate, vigorous and MVPA). The threshold for identifying non-wear time will be based on reviewing the acceleration data and will follow a process to correct for diurnal bias similar

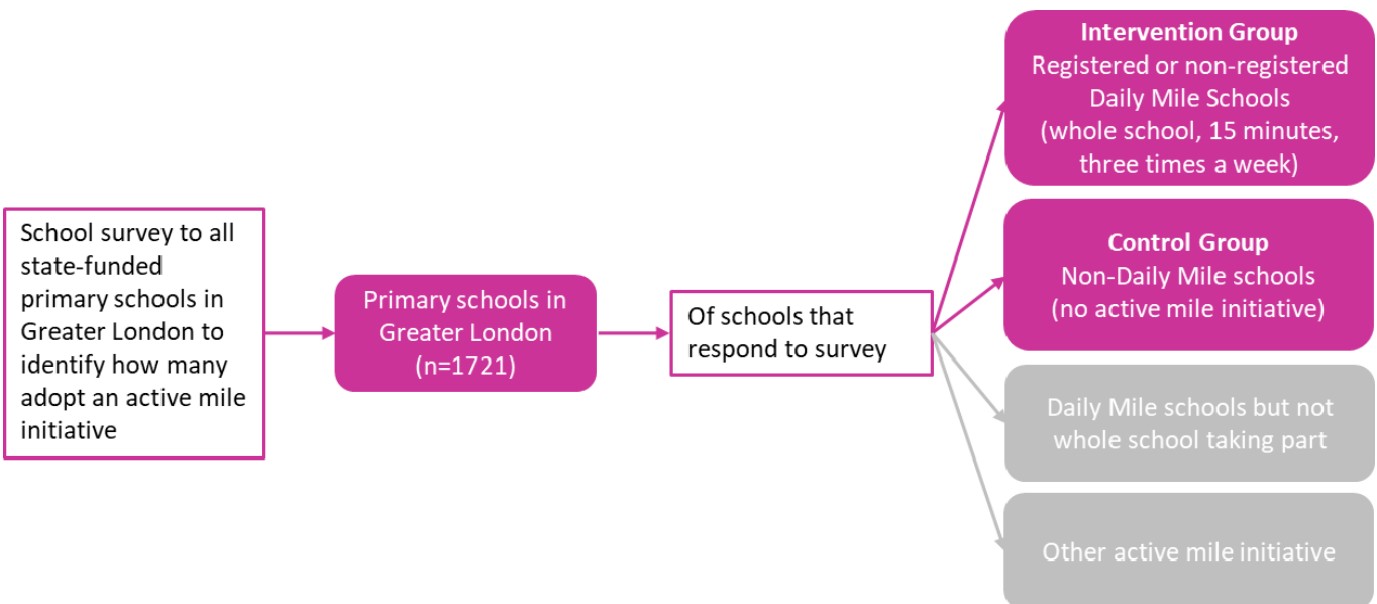

**Figure 2** Identifying schools to be included in the iMprOVE Study.

to that used in previous studies.[26 27] As the use and analysis of wrist-worn accelerometer data in younger age groups is still developing,[28 29] we will review and adapt our process to reflect the updated knowledge and best practice at the time of first analysis. We will use segmented analysis to identify MVPA during TDM activities as reported by the class teacher, and to consider compensatory effects, we all assess minutes of MVPA during school hours and outside school hours and on weekdays overall and at the weekend.

The Strengths and Difficulties Questionnaire (SDQ), a measure of child mental health,[30] is a brief behavioural screening questionnaire about children and young people aged from 2 years to 17 years. The SDQ has five subscales each including five items that assess emotional symptoms, conduct problems, hyperactivity, peer relationship problems and prosocial behaviour. It has been extensively validated[31] and is one of the most widely and internationally used measures for assessing children's mental health.[32] Parent/carers and teachers will complete the SDQ about the child.

Educational attainment will be measured by teacher ratings of the child's reading, writing and math ability on a 5-point Likert Scale ranging from 1 (below expected levels) to 5 (above expected levels).[20] Ratings will be based on the UK's National Curriculum's 'age related expectations' which are the standard expectations defined by threshold descriptors indicating what pupils should be able to achieve by the end of primary school.[33]

### Secondary outcomes

We will assess children's self-reported (or completed by proxy) intensity of physical activity using the Children's Physical Activity Questionnaire (C-PAQ).[34] Collecting self-reported physical activity alongside accelerometer data will enable us to validate self-reported data with our objective measure of physical activity.

The Child Health Utility 9D (CHU-9D) will measure children's health-related quality of life (HRQoL).[35] The CHU-9D assesses how children are feeling 'today' by addressing nine items (worry, sadness, pain, tiredness, annoyance, problems with schoolwork, their daily routine and their ability to join in with activities). It is suitable for children aged from 7 years to 11 years and can be completed by proxy for younger children.[36] The CHU-9D has good psychometric performance and shows potential as a valid and reliable instrument for measuring HRQoL.[37–39]

At baseline (children in school year 1) and at the first follow-up (children in year 2) the C-PAQ and CHU-9D will be completed by proxy about the child by the parent/carer. From school year 3 until the end of primary school, children will be encouraged to complete their own questionnaires with the help of their parent/carer (online supplemental table 2).

We will measure children's height and weight from which we will calculate each child's BMI as weight/height$^2$ (kg/m$^2$) and standardised by age and sex.

### Covariates

Parental physical and mental health is a strong predictor of child functioning.[40] We will therefore collect parental self-reported physical activity, mental health, well-being and general health. The International Physical Activity Questionnaire-Short Form[41] will measure time spent in different intensities of physical activity during the last 7 days. We will use the Warwick and Edinburgh Mental Wellbeing Scale which includes 14 items of feeling and functioning aspects of well-being to measure parental mental health.[42]

We will collect sociodemographic data on children's date of birth, sex and ethnic group and parental sociodemographic data including age, sex, ethnic group, highest level of qualification and employment status. These data

will be collected through questionnaires completed by the parent/carer. For the covariates of age/date of birth, sex and ethnic group of the children and parent/carer, the questions will only be included at baseline. Parent/carer qualification and employment data will be asked at each follow-up stage as these could change throughout the study period.

School characteristics including school size (ie, overall pupil numbers), class size, ethnic composition, area level of deprivation and number of pupils eligible for free school meals will be obtained from publicly available data provided by the Department of Education. School participation in TDM will be sought at the start of each follow-up data collection.

## Data collection

We will email a link to the questionnaires to parent/carers and class teachers. Paper versions will be available on request. Parents will complete the child questionnaires by proxy.

The iMprOVE research team will arrange a convenient time to visit the school to take anthropometry measures of the children and issue and fit them with the accelerometer. Verbal instructions of wearing the monitor will be given along with an information pack including details of wearing the monitor correctly, how long to wear and when and how to return the monitors. Incentives (shopping voucher) will be offered to the child to encourage successful completion of all their corresponding data collection.

All measures taken at baseline will be repeated at each follow-up. From Wave 3 onwards, the children (now in year group 3 and aged 7–8 years) will be encouraged to complete their own questionnaires with help from their parent/carer. Incentives for the children after each wave of data collection will be offered to encourage families to continue taking part.

## Data collection time points

There will be one data collection in each school year. Data collection for baseline (Wave 1) will begin from January 2021 (allowing for regular TDM activity to take place if implemented by the school) and will continue until we have met our sample size targets. Control group participants will be assessed simultaneously to the intervention group (figure 1).

For consistency and potential seasonal variation in activity levels, we aim to carry out follow-up assessments during the same month as baseline assessments in each year until the end of data collection in 2026 (figure 1).

## Sample size

Our sample size is based on the primary outcome of mean MVPA (minutes) per day allowing for the clustering of children in the same school and the primary analysis adjusting for baseline minutes of MVPA. Based on our previous work, there is an average of 46 children per state-funded primary school in Greater London.[11]

We conducted a sample size calculation (using STATA V.15.1) that produced a total sample of 3533 children (approximately 1767 children in the intervention and control groups) across 77 schools to give the study 90% power to detect a 5.5 min increase in the mean daily average MVPA minutes between children in TDM schools compared with non-TDM schools, assuming a mean daily average MVPA of 53 min (SD 22) in the control group based on the study by Chesham et al.[15] This has been inflated to allow for 60% attrition across the follow-up period and assumes a two-sided alpha equal to 0.05. To allow for the analyses to include clustering at school level, we have applied an intraclass correlation coefficient of 0.05 to our calculation (online supplemental table 3). An increase of just over 5 min of MVPA per day equal to a 10% difference would be clinically significant to increase physical activity among the proportion of children who participate in less than an average of 30 min per day.

## Statistical analyses

For every wave of data collection, the main analysis will treat the outcomes as continuous variables. This will enable us to understand the impact of TDM in any change observed in physical activity, mental health and educational attainment for each child through each successive year of primary school.

We will use a linear multilevel mixed effect regression model to estimate the mean difference in MVPA values between TDM schools and non-TDM schools across follow-up and at each follow-up wave. Children will be included as a random intercept with fixed effects for wave, intervention group, wave-by-intervention group interaction and baseline (Wave 1) MVPA. School will also be included in the model as a random effect and adjustments for covariates including participation in TDM and sociodemographic variables will be included. Baseline adjusted group differences will be reported with a 95% CI for each wave. An additional model without wave-by-intervention interaction will be fitted to provide an estimate of the overall intervention effect across follow-up.

Based on previous studies,[40 43] we expect data from children to vary throughout each wave of data collection as compliance in the youngest age groups (5–6 years) may differ compared with when they are measured later (aged 10–11 years). Decisions on valid accelerometer data to include in the analysis will be made when reviewing the data and based on previous literature.

Linear multilevel mixed effect regression models as described for MVPA will be repeated for the outcomes of mental health and educational attainment.

### Subgroup analysis

To address inequalities, we will examine MVPA of children in TDM and non-TDM schools according to area level deprivation (a measure of poverty) using the Index of Multiple Deprivation.[44]

## Sensitivity analyses

We anticipate that during the study period, schools that are assigned to intervention group may stop TDM and schools assigned to the control group may take up TDM. We will conduct a series of sensitivity analyses to explore TDM impacts in children who engage in TDM in some year groups only. We will also consider seasonal effects on implementation of TDM and MVPA overall dependent on the month children are assessed. In addition, missing data for MVPA in the follow-up waves will be imputed and included separately to a complete case analysis.

All analyses will be carried out using STATA V.15 (StataCorp, Texas, USA).

## Patient and public involvement

We will conduct focus group workshops with children in the intervention group to ask about their views on taking part in TDM activities and their physical activity inside and outside of school. A separate focus group workshop will be conducted with the control group to obtain their views on physical activity inside and outside of school. The workshops will take place once a year during each year of data collection.

We have consulted professional and public representatives within the management structure (TDM Research Advisory Group including members from TDM Foundation, academic researchers and representatives from Sport England, London Marathon and London Sport and patient and public involvement including parents and children who have been approached for piloting questionnaires and assessments) throughout the design of the study and who will continue to be consulted throughout the duration of the study. Advice on the design of study materials (eg, study information sheets and questionnaires), participant engagement and how to retain participants in the study will be sought throughout the study period. Questionnaires will be piloted with parents and children through workshops.

## DISCUSSION

Currently, physical activity has fallen to lowest rates following the COVID-19 pandemic lockdowns and restrictions. Global policies are focusing on safe ways that will allow children to adapt to primary school life that allows improved health behaviours aimed at increasing physical activity. Active mile initiatives in school settings are seen as part of the solution to rebuilding and embedding health behaviours in early life.

The iMprOVE Study will be the most comprehensive evaluation of an active mile initiative and the first longitudinal study to capture the impact of participation over primary school life.

Key strengths of the iMprOVE Study are its size, the assessment of an intervention in a real world setting and individual level data and longitudinal follow-up throughout primary school life. Furthermore, the objective measure of physical activity using accelerometers will provide more accurate estimates of daily minutes of MVPA than parent-reported physical activity in questionnaires, minimising potential participant bias of overestimating or underestimating self-reported time spent being active.[45] The inclusion of schools from a diverse multiethnic population and urban setting enables us to examine differential impacts of TDM among different sociodemographic groups. However, there are a number of important limitations to consider. Since iMprOVE is a quasi-experimental cohort study and not randomised, there is likely to be an unequal number of participants in the intervention and control groups. We therefore expect some selection bias in schools that agree to participate in the study and as with previous cohort studies, there may be under-representation of school populations from disadvantaged areas.

Given the popularity of TDM and its widescale adoption across the UK and globally, the results of this study will be of interest to parents, teachers and public health policymakers. The findings will determine the potential of TDM and other similar active mile initiatives, to improve the health, well-being and educational attainment for all children in primary schools and inform national strategies to reduce inequalities in child health.

## Ethics and dissemination

Full ethical approval has been granted by Imperial College Research Ethics Committee (ICREC), reference 20IC6127.

Wider dissemination of key findings to the public will be through research networks, social (Facebook, Twitter), print (newspapers) and media broadcasts (news outlets). We will engage with study participants through community engagement activities, presentations and newsletters to schools, teachers, parents and children taking part in the study. We will work with policy-makers nationally to ensure direct application and impact of our work.

## Project steering

The management structure of this study includes the following groups:
- ► Investigators (principal investigator and lead researcher).
- ► Research support team (collaborators).
- ► Project steering group (advisors).

**Acknowledgements** The authors are grateful for support from the National Institute for Health Research School for Public Health Research (NIHR SPHR) and the NIHR North West London Applied Research Collaboration (NW London ARC). The NIHR School for Public Health Research is a partnership between the Universities of: Sheffield; Bristol; Cambridge; Imperial; and University College London; The London School of Hygiene and Tropical Medicine; LiLaC—a collaboration between the Universities of Liverpool and Lancaster; and Fuse—the Centre for Translational Research in Public Health, a collaboration between Newcastle, Durham, Northumbria, Sunderland and Teesside Universities. The Department of Primary Care and Public Health is grateful for support from the NIHR Biomedical Research Centre funding scheme, the NIHR School for Public Health Research and the NIHR North West London Applied Research Collaboration. This report is independent research supported by the National Institute for Health Research Collaboration Northwest London. The views expressed in this publication are those of the authors

and not necessarily those of the National Institute for Health Research or the Department of Health and Social Care, the Department for Education, or The Daily Mile Foundation. The authors thank The Daily Mile (TDM) Foundation for funding and support.

**Contributors** SS wrote the original application and obtained funding for the study. BR and SS conceptualised the cohort study and wrote the ethics application and gained ethical approval from Imperial College Research Ethics Committee. BR, SS, AC, EvS, DSH and RMV designed the study. BR and RP developed the statistical analysis plan. BR wrote the methodology, developed all the study materials and produced the first draft of this manuscript. BR, SS, AC, EvS, DSH, RMV and TV contributed to redrafts.

**Funding** This work is funded by The Daily Mile (TDM) Foundation supported by INEOS. BR is funded by TDM Foundation. SS is supported by TDM Foundation, the National Institute for Health Research (NIHR) School for Public Health Research (SPHR) and the NIHR North West London Applied Research Collaboration (NW London ARC). TV is funded by an NIHR SPHR PhD Studentship (Grant Reference Number PD-SPH-2015-10055). DSH is part of the NIHR SPHR and NW London ARC and is supported by the Department for Education. EvS is supported by the Medical Research Council (MRC) (Grant MC_UU_12015/7). The work by EvS is undertaken under the auspices of the Centre for Diet and Activity Research, where funding from Cancer Research UK, the British Heart Foundation, the Economic and Social Research Council, the MRC, the NIHR and the Wellcome Trust, under the auspices of the UK Clinical Research Collaboration, is gratefully acknowledged (087636/Z/08/Z; ES/G0007462/1; MR/K023187/1).

**Competing interests** SS and BR received funding from The Daily Mile Foundation. SS, BR, AC and TV are members of The Daily Mile Research Advisory Group.

**Patient consent for publication** Not required.

**Provenance and peer review** Not commissioned; externally peer reviewed.

**ORCID iDs**
Bina Ram http://orcid.org/0000-0003-0023-1573
Anna Chalkley http://orcid.org/0000-0002-1163-6210
Rachel Phillips http://orcid.org/0000-0002-3634-7845
Tishya Venkatraman http://orcid.org/0000-0001-6171-2384
Russell M Viner http://orcid.org/0000-0003-3047-2247
Sonia Saxena http://orcid.org/0000-0003-3787-2083

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
