## [Reviewer comments · BMJ Open]

ARTICLE DETAILS

TITLE (PROVISIONAL)	Impact of The Daily Mile™ on children's physical and mental health and educational attainment in primary schools; iMprOVE cohort study protocol
AUTHORS	Ram, Bina; Chalkley, Anna; van Sluijs, Esther; Phillips, Rachel; Venkatraman, Tishya; Hargreaves, Dougal; Viner, Russell; Saxena, Sonia

VERSION 1 – REVIEW

REVIEWER	Hanna Nalecz Institute of Mather and Child, Poland
REVIEW RETURNED	09-Dec-2020

GENERAL COMMENTS	Dear Authors, It was my great pleasure to review your protocol. This is a well-written, substantive and meeting all the criteria of a good protocol, description of the planned evaluation of The Daily Mile initiative. A few small remarks that may improve protocol quality are bolded in the text below. (in case of the editing surprises, please see the attached pdf. file) The introduction is informative, the authors elaborate all aspects appearing in the project, which they support in the quoted, fresh and appropriate, literature. Aims and research questions are very clear, and proposed methodology corresponds to aims and research questions. The study design is fully and clearly described, but biased, however the Authors are aware of this fact and explain the possible impact. The study design, intervention and procedure are clearly described. For the planned 6-waves-study (January 2021-2026) the Authors obtained an Ethical approval, and all needed informed consent are design to receive through the study procedure. Supplementary materials are clear and valuable for the reader. Figure 2 might be later used as a flow chart after implementing the "n" at the different stages of sampling. In the description of outcomes and measures (also Table 2) the Authors listed wrist-worn accelerometers as an objective measurement of MVPA. It would be desirable to define exemplary devices here. Timeline and procedure of the data collection are clearly described, including the definition of start and endpoint of the study. I am impressed with the accuracy of the description of the rationale for the sample size and the planned statistical analyses.
--

	It is very valuable that Authors included in the procedure consultations with the community and experts at the research and tools design stage and in its description, but I suggest changing the title of this section to Local and public involvement to avoid using "patient". In the discussion, the authors showed that they understand the broader context of their research and that they are aware of the bias resulting from the sampling method and the applied tools. The source of the project financing was indicated, it would be good to supplement the protocol with the Authors' conflict of interest declarations. I wish the Authors good luck during the implementation of the project, and I am looking forward to reading the publication. Kind regards, Reviewer
--	--

REVIEWER	Dr Jade Lynne Morris King's College London
REVIEW RETURNED	22-Dec-2020

GENERAL COMMENTS	Overall I think this is an exciting project and the protocol has been well written. Some of the measurements require some more detail to ensure the protocol could be replicated. Ram et al., 2020. Impact of The Daily Mile™ on children's physical and mental health and educational attainment in primary schools; iMprOVE cohort study protocol Abstract  • Line 47-50: Perhaps only a preference, but I would suggest splitting this sentence up about the survey and -non-random allocation. At the moment it is housing a lot of information and may be better split up to show the two steps involved in this process. Perhaps "A survey will be distributed to all state-funded primary schools in Greater London to identify participation in The Daily Mile. The survey responses will be used for non-random allocation to either the intervention (Daily Mile schools) or control (non-Daily Mile schools)". • Lines 52-55: Methods and analyses – can you add the frequency of data collection. Introduction  • Line 82 and 83: You mention 'sport and physical activity' when in fact the CMO guidelines are for physical activity which is an umbrella term encompassing sport, P.E., active travel, play etc. It might be clearer to stick to physical activity here and remove the Sport England reference from this sentence. • Line 90-91: Perhaps it might be worth adding in the sentence about TDM launch, adding in it was launched by a headteacher aiming to increase aerobic fitness levels. "TDM was launched in 2012 in Scotland by a Head Teacher looking to increase the aerobic fitness levels of her pupils (REF)." • Line 95: "initiatives and have" – remove the and. • Line 95: Does TDM have the potential to level health inequalities, or just have the potential to help shorten the gap? I haven't thoroughly inspected the reference; however, I think levelling health inequalities seems to be a big claim for just implementing
--

TDM. There are more factors at play than just physical activity provision at school. For example, physical activity (or lack of) and diet outside of the school day are additional factors that are likely to impact health inequalities.

- Lines 97-106: This is a nice summary of the existing evidence on TDM. However, there isn't any evidence on physical activity levels which is meant to your main measure for this study. I would suggest including some of the evidence that has come out on the amount of physical activity that takes place during TDM compared to staying in the classroom and any evidence on the change of MVPA over time. See the Active Mile Briefing paper for some references on this. I think by adding this in the next paragraph, 107-109 would fit better. You can then say something along the lines of this "while there are acute assessments suggesting children on average accumulate 10 mins MVPA during TDM and 1-year longitudinal assessments of TDM also suggesting a 10 minutes increase in daily MVPA, the evidence base still lacks an assessment of the impact and sustainability of TDM throughout primary school life."

Method and analyses

- Line 135: you state TDM is running or jogging but I believe TDM guidance states running, jogging or walking so it might be nice to add walking in here. I notice in Line 145-146 you mention the consensus that children should be active 'running or jogging'. I would argue walking is also 'active' unless it needs to be of a moderate intensity. Perhaps make this clearer then if 'walking' was not deemed acceptable.

- Line 136: Start a new sentence at 'in addition'. "TDM is in addition to P.E. lessons and scheduled break times."

- Line 143: You have not abbreviated TDM in this sentence

- Line 148 onwards: This sections seems ambiguous and I have several questions around the process of the survey and recruitment that should add clarity:

- o How many times will you contact schools with the survey?

- o Will this be via email only? Will you call them up?

- o Who are you asking to fill out the survey? Only senior leadership teams or any teacher in the school?

- o What is the time frame for getting this information before you cut them off from not being included in the study? I noticed there is some timeline information in Fig 1. Perhaps add in number of times contacted into this, or write it in?

- Lines 170 onwards: the main outcome measure of physical activity using accelerometers required more details:

- o If you collect data from 9am until 3pm what about schools that start earlier or finish later? This would be better stated 'data will be collected during the school day which typically runs 9-3pm, but this will be adjusted per school'?

- o Will you conduct segmented analysis on the school day? It would seem wasteful not to get a timetable of the school day and identify when TDM takes place for each class – this means you can look at the direct contribution of MVPA from TDM within TDM schools and not just compare daily in-school MVPA with control schools.

- o It seems a bit confusing you then say you will capture outside of school and weekend. Perhaps state, children will wear accelerometers for 7 days and segmented analysis will look at specifically at the school day...

- o What models of accelerometers are you using? How will you set up and analyse the accelerometers? E.g., raw data, counts data,

	what epoch length, what about non-wear time? Will they take them off to sleep?  o Will you collect an additional day to discount the first day for reactivity? If not, will you assess if day one differs from the rest of the week? • Line 174 – for the SDQ, on the supplement file you say this is to be completed by the teacher and parents. Can you clarify if they are filling it out for themselves, or if they will just be helping the children fill it out? I think the supplement table may need clarifying as this isn't very clear and I presume it is the children that are filling this out. • Line 188: you say the C-PAQ will allow you to validate the objectively assessed PA data. I would argue the PA data from objectively assessed, recommended accelerometers would actually validate the C-PAQ self-report measures that are subject to bias. Unless you are using a novel, not previously used method or analysing the accelerometer data. I would consider re-wording. • Can you clarify if the IPAQ is being completed by parents for children's PA levels or for the parents PA levels. Can you justify why you are using the IPAQ and the C-PAQ and clarify the use of these. (It might make sense to include who is filling out what when you discuss each measurement). • Lines 204-206: How will you collect this? Asking the school, children, parents or a mix? • Lines 236 onwards: Just to make this a bit clearer, can you add that you have done an 'a priori power calculation' just to really emphasise the work gone on here to ensure the sample is powered for statistical change in MVPA. Might also be worth adding what software you used to run the calculation. A few other things that came to mind you might want to consider:  • Have you considered collecting MVPA levels at two time points during the school year? Perhaps collecting MVPA data in both the winter and summer. There is substantial existing evidence to suggest MVPA levels differ across the seasons. In the UK, we get a lot of rain during the winter and it might also be worth capturing the weather during data collection weeks – does this stop children going out and completing TDM. Does this decrease engagement? • Have you also considered speaking to the children at each time point to see understand their PA behaviours; how they may change over the. Years and how they feel about TDM. Do they get bored of this, doing it every day? Will there be differences in schools that enforce it every day irrespective of the weather compared to schools that do it sporadically. • Will you record if the class teacher changes. Will you do any observations of the teachers and how they actually implement TDM – this is likely to be different within each class. What impact might this have? E.g., a teacher that runs with them compared to a teacher that takes an umbrella out in the rain and stands there and shouts at the children to keep going but doesn't get involved. • Will you record if the schools engaging in any other whole school approaches to PA? This might be important in case there is a shift away from TDM to something else. Overall, I think this is a really exciting project and there is so much to explore. So, if there are other variables you can capture it will only make the data richer and more insightful for understanding the sustainability of TDM and the impact this has on children's relationships with physical activity.
--	---

VERSION 1 – AUTHOR RESPONSE

Reviewer 1: Dr Hanna Nalecz, Instytut Matki i Dziecka

Comment	Response	Amendment/Revision
Comment #1: In the description of outcomes and measures (also Table 2) the Authors listed wrist-worn accelerometers as an objective measurement of MVPA. It would be desirable to define exemplary devices here.	We have now defined the devices that we will be using.	Page 6, line 194-195 “...measured using the GENEActiv (Activinsights, Cambridge, UK) wrist-worn accelerometers.”
Comment #2: It is very valuable that Authors included in the procedure consultations with the community and experts at the research and tools design stage and in its description, but I suggest changing the title of this section to Local and public involvement to avoid using "patient".	We have checked the BMJ Open website (https://bmjopen.bmj.com/pages/authors/) and the heading we have used is as requested by the journal: “To support co-production of research we request that authors provide a Patient and Public Involvement statement in the methods section of their papers, under the subheading ‘Patient and public involvement’.”	
Comment #3 The source of the project financing was indicated, it would be good to supplement the protocol with the Authors' conflict of interest declarations	The competing interests is declared on the manuscript.	Page 13, lines 414-415 (as in manuscript, no changes made) SS and BR received funding from The Daily Mile Foundation. SS, BR, AC and TV are members of The Daily Mile Research Advisory Group.

Abstract Comment #1: Line 47-50: Perhaps only a preference, but I would suggest splitting this sentence up about the survey and -non-random allocation. At the moment it is housing a lot of information and may be better split up to show the two steps involved in this process. Perhaps “A survey will be distributed to all state-funded primary schools in Greater London to identify participation in The Daily Mile. The survey responses will be used for non-random allocation to either the intervention (Daily Mile schools) or control (non-Daily Mile schools)”.	We have split the sentence as requested.	Page 2, lines 47-50 “We will send a survey to all state-funded primary schools in Greater London to identify participation in The Daily Mile. The survey responses will be used for non-random allocation to either the intervention group (Daily Mile schools), or to the control group (non-Daily Mile schools).”
Comment #2: Lines 52-55: Methods and analyses – can you add the frequency of data collection.	We have added the frequency of data collection to the Methods and analysis section of the Abstract.	Page 2, lines 50-53 “We aim to recruit 3,533 Year 1 children (aged 5-6 years) from 77 primary schools and follow them up annually until the end of their primary school years. Data collection taking place at baseline (children in school Year 1), and each primary school year thereafter, includes objectively measured moderate to vigorous physical activity (MVPA)...”,
Introduction Comment #3: Line 82 and 83: You mention ‘sport and physical activity’ when in fact the CMO guidelines are for	We have removed the reference for Sport England (reference 5 in	Page 4, line 85 “...for an average of 60

physical activity which is an umbrella term encompassing sport, P.E., active travel, play etc. It might be clearer to stick to physical activity here and remove the Sport England reference from this sentence.	manuscript)	minutes or more every day.[1]”
Comment #4: Line 90-91: Perhaps it might be worth adding in the sentence about TDM launch, adding in it was launched by a headteacher aiming to increase aerobic fitness levels. “TDM was launched in 2012 in Scotland by a Head Teacher looking to increase the aerobic fitness levels of her pupils (REF).”	We have added information about the TDM launch.	Page 4, lines 90-91 “The Daily Mile™ (TDM) is a school-based active mile initiative that was launched in 2012 by a head teacher in Scotland, UK, who wanted to increase the fitness of her pupils.[2]”
Comment #5: Line 95: “initiatives and have” – remove the and.	We have removed the ‘and’ from this sentence.	Page 4, line 100 “...school-based active mile initiatives have the potential to...”
Comment #6: Line 95: Does TDM have the potential to level health inequalities, or just have the potential to help shorten the gap? I haven’t thoroughly inspected the reference; however, I think levelling health inequalities seems to be a big claim for just implementing TDM. There are more factors at play than just physical activity provision at school. For example, physical activity (or lack of) and diet outside of the school day are additional	We have amended the sentence accordingly.	Page 4, lines 100-102 “...school-based active mile initiatives have the potential to reach all children, including those living in poverty, which may help to reduce the gap in physical activity among children.[3]”

factors that are likely to impact health inequalities.		
Comment #7: Lines 97-106: This is a nice summary of the existing evidence on TDM. However, there isn't any evidence on physical activity levels which is meant to your main measure for this study. I would suggest including some of the evidence that has come out on the amount of physical activity that takes place during TDM compared to staying in the classroom and any evidence on the change of MVPA over time. See the Active Mile Briefing paper for some references on this. I think by adding this in the next paragraph, 107-109 would fit better. You can then say something along the lines of this "while there are acute assessments suggesting children on average accumulate 10 mins MVPA during TDM and 1-year longitudinal assessments of TDM also suggesting a 10 minutes increase in daily MVPA, the evidence base still lacks an assessment of the impact and sustainability of TDM throughout primary school life."	We have now included evidence on physical activity that takes place during TDM.	Page 4, lines 105-112 "The evolving evidence base of TDM on physical activity is promising: a study by Chesham et al (2018) showed an increase in children's moderate to vigorous physical activity (MVPA) by ~9 minutes,[4] and a study by Morris et al (2020) demonstrated that during one active mile lesson, children accumulated between 5 and 15 minutes of MVPA.[5] Whilst there are acute assessments suggesting that children accumulate around an average of 10 minutes of MVPA during TDM, there is limited evidence of its impact on children's mental health[6, 7] and inconsistencies on benefits to academic performance.[5, 8]" Page 4, lines 116-117 "The evidence base still lacks an assessment of the impact and sustainability of TDM throughout primary school life."
Method and analyses Comment #8: Line 135: you state TDM is running or jogging but I believe TDM guidance states running, jogging or walking so it might be nice to add walking in here. I/	We have checked TDM's website to confirm whether they include walking as part of their initiative (https://thedailymile.co.uk/about/).	Page 6, lines 162-163 "...(2) children are running or jogging for 15 minutes..."

notice in Line145-146 you mention the consensus that children should be active ‘running or jogging’. I would argue walking is also ‘active’ unless it needs to be of a moderate intensity. Perhaps make this clearer then if ‘walking’ was not deemed acceptable.	They state: “The Daily Mile is a social physical activity, with children running or jogging – at their own pace – in the fresh air with friends. Children can occasionally walk to catch their breath, if necessary, but should aim to run or jog for the full 15 minutes.” As running or jogging is encouraged for the full 15 minutes, we did not feel that walking should be included. Our primary outcome is moderate to vigorous physical activity which is more likely to be accumulated during running and jogging. We have made this clearer in the manuscript.	
Comment #9: Line 136: Start a new sentence at ‘in addition’. “TDM is in addition to P.E. lessons and scheduled break times.”	We have split this sentence.	Page 5, line 153 “...during the school day. TDM is in addition to curricular lessons...”
Comment #10: Line 143: You have not abbreviated TDM in this sentence	This has been amended.	Page 6, line 160 “...and members from TDM Foundation.”
Comment #11: Line 148 onwards: This sections seems ambiguous and I have several questions around the process of the survey and recruitment that should add clarity: a) How many times will you contact schools with the survey?	We have clarified this section as requested. In summary: a) The school survey will take place once to identify which schools to assign to the intervention and control groups at baseline.	Page 6, Line 166 “To identify which schools to assign to the intervention

b) Will this be via email only? Will you call them up? c) Who are you asking to fill out the survey? Only senior leadership teams or any teacher in the school? d) What is the time frame for getting this information before you cut them off from not being included in the study? I noticed there is some timeline information in Fig 1. Perhaps add in number of times contacted into this, or write it in?	b) We will initially chase up by email followed by a phone call. c) Any teacher that is best placed to answer questions on the school's involvement in TDM can complete the survey. We have included a question in the school survey that asks for the role of the person completing the survey. d) The time frame to collect this information will be from September to December within which time we hope to reach our recruitment targets.	and control groups at baseline, we will send a..." Page 6, Lines 169-172 "Any teacher who is best placed to answer questions about their school's involvement with TDM can complete the survey. Schools that do not respond will be contacted by email two weeks after initial contact with a further email one week later. If required, we will conduct a further follow-up by phone to encourage survey completion two weeks after the second email."
Comment #12: Lines 170 onwards: the main outcome measure of physical activity using accelerometers required more details: a) If you collect data from 9am until 3pm what about schools that start earlier or finish later? This would be better stated 'data will be collected during the school day which typically runs 9-3pm, but this will be adjusted per school'?" b) Will you conduct segmented analysis on the school day? It would seem wasteful not to get a	We have added more detail to clarify the points raised by the reviewer. In summary: a) We will match the children to the start and end times of the school they attend (authors have successfully implemented this approach on previous studies). b) We agree that it is important to identify timing of TDM in the school day to assess the direct contribution to physical activity.	Page 6, lines 192-193 "...(we will match individual children to the start and end times of their school as reported by the teacher)." Page 5, Lines 135-136 "How much MVPA is

timetable of the school day and identify when TDM takes place for each class – this means you can look at the direct contribution of MVPA from TDM within TDM schools and not just compare daily in-school MVPA with control schools. c) It seems a bit confusing you then say you will capture outside of school and weekend. Perhaps state, children will wear accelerometers for 7 days and segmented analysis will look at specifically at the school day... d) What models of accelerometers are you using? How will you set up and analyse the accelerometers? E.g., raw data, counts data, what epoch length, what about non-wear time? Will they take them off to sleep? e) Will you collect an additional day to discount the first day for	It has now been included. c) We have revised this sentence. d) Please see response to Reviewer 1 Comment 1 about the model of accelerometer. The use of wrist worn accelerometry in younger age groups is still uncommon and we are therefore unable to provide definitive plans for processing and data reduction until we can review the accelerometer data. We have added further detail on wear instructions and our current plans for data processing and reduction while acknowledging that this will reflect knowledge and best practice that may still emerge. e) We have added detail on the number of days we will collect data. Please see revisions made on manuscript in response to comment #12d above where we have now specified that the accelerometers will be initialised for 9 days to obtain seven full days of wear.	attributed to doing TDM and how much does this contribute to children’s overall physical activity?” Pages 7, lines 204-205 “We will use segmented analysis to identify MVPA during TDM activities as reported by the class teacher...” Page 7, lines 205-207 “...and to consider compensatory effects, we will assess minutes of MVPA during school hours and outside school hours, and on weekdays overall and at the weekend.” Page 6, lines 195-196 “The GENEActiv accelerometer is worn 24/7 including when in water and during sleep.” Pages 6-7, lines 196-204 “Children will be instructed to wear the monitors for 24 hours on each day of a consecutive 7-day wear period. The accelerometers will be initialised for 9 days to obtain 7 full days of wear (excluding the date of issue and day of collecting the accelerometers). We will use raw acceleration data to classify wear time and assess time spent in activity intensities (including moderate, vigorous and
--	---	--

reactivity? If not, will you assess if day one differs from the rest of the week?		moderate to vigorous physical activity). The threshold for identifying non-wear time will be based on reviewing the acceleration data and will follow a process to correct for diurnal bias similar to that used in previous studies [9,10]. As the use and analysis of wrist-worn accelerometer data in younger age groups is still developing [11,12], we will review and adapt our process to reflect the updated knowledge and best practice at the time of first analysis."
Comment #13: Line 174 – for the SDQ, on the supplement file you say this is to be completed by the teacher and parents. Can you clarify if they are filling it out for themselves, or if they will just be helping the children fill it out? I think the supplement table may need clarifying as this isn't very clear and I presume it is the children that are filling this out.	We have added clarification on the completion of the SDQ. We have also amended Supplemental Table 2 for clarification (please see Appendix 1 in this document).	Page 7, line 214 "Parent/Carers and teachers will complete the SDQ about the child." Supplemental Table 2 included as Appendix 1 in this document.
Comment #14: Line 188: you say the C-PAQ will allow you to validate the objectively assessed PA data. I would argue the PA data from objectively assessed, recommended accelerometers would actually validate the C-PAQ self-report measures that are subject to bias. Unless you are using a novel, not previously used method or	We have re-worded this sentence.	Page 7, Lines 223-224 "...will enable us to validate self-reported data with our objective measure of physical activity."

analysing the accelerometer data. I would consider re-wording.		
Comment #15: Can you clarify if the IPAQ is being completed by parents for children's PA levels or for the parents PA levels. Can you justify why you are using the IPAQ and the C-PAQ and clarify the use of these. (It might make sense to include who is filling out what when you discuss each measurement).	Please see response to Comment #13 (Reviewer 2) where we have amended Supplemental Table 2 to clarify who is completing which questionnaire. Under the section 'Covariates', we had included an explanation for collecting IPAQ. Under the section 'Secondary outcomes', we had included an explanation for collecting CPAQ. Please see amendment to Comment 14 (Reviewer 2). We have now included who is filling in which questionnaire.	Please see Appendix 1. Page 7, lines 238-239 (as in manuscript, no changes made) Parental physical and mental health is a strong predictor of child functioning. We will therefore collect parental self-reported physical activity, mental health... Page 7, lines 231-234 "At baseline (children in school year 1) and at the first follow-up (children in year 2) the C-PAQ and CHU-9D will be completed by proxy about the child by the parent/carer. From school year 3 until the end of primary school, children will be encouraged to complete their own questionnaires with the help of their parent/carer (Supplemental Table 2)."
Comment #16: Lines 204-206: How will you collect this? Asking the school, children, parents or a mix?	Supplemental Table 2 (Appendix 1) now clarifies how we will collect socio-demographic data.	Page 8, Lines 246-249 "These data will be collected through questionnaires completed by the parent/carer. For the

		covariates of age/date of birth, sex and ethnic group of the children and parent/carer, the questions will only be included at baseline. Parent/Carer qualification and employment data will be asked at each follow-up stage as these could change throughout the study period.”
Comment #17: Lines 236 onwards: Just to make this a bit clearer, can you add that you have done an ‘a priori power calculation’ just to really emphasise the work gone on here to ensure the sample is powered for statistical change in MVPA. Might also be worth adding what software you used to run the calculation.	As this is or study protocol, our sample size calculations were done prior to any data collection. To make this clearer, the authors agreed that we should add ‘sample size calculation’ to the text.	Page 9, Line 279 “We conducted a sample size calculation (using Stata v15.1) that produced a total sample of...”
Comment #18: A few other things that came to mind you might want to consider: Have you considered collecting MVPA levels at two time points during the school year? Perhaps collecting MVPA data in both the winter and summer. There is substantial existing evidence to suggest MVPA levels differ across the seasons. In the UK, we get a lot of rain during the winter and it might also be worth capturing the weather during data collection weeks – does this stop children going out and completing TDM. Does this decrease engagement?	We have allowed for one assessment per child in each academic year given our resources, sample size, and the need to allow sufficient time for recruitment and for schools to implement TDM before assessments. However, we will be including the month in which assessments take place in our analysis models that was specified in the manuscript which will allow us to assess for seasonal variations. All follow-ups will take place during the same month in which the baseline measure took place.	Page 9, lines 313-316 (as in manuscript, no changes made) We will conduct a series of sensitivity analyses to explore TDM impacts in children who engage in TDM in some year groups only. We will also consider seasonal effects on implementation of TDM and MVPA overall dependent on the month children are assessed.
Comment #19:		

Have you also considered speaking to the children at each time point to see understand their PA behaviours; how they may change over the. Years and how they feel about TDM. Do they get bored of this, doing it every day? Will there be differences in schools that enforce it every day irrespective of the weather compared to schools that do it sporadically.	We have embedded focus group workshops into our plan of investigation to specifically ask about children's views on TDM in the intervention group, and about general physical activity in both the intervention and control groups. We plan to carry out these workshops once in every year of data collection for the intervention and control groups separately. We have now added this information to the manuscript.	Page 10, lines 320-324 “We will conduct focus group workshops with children in the intervention group to ask about their views on taking part in TDM activities and their physical activity during and outside the school day. A separate focus group workshop will be conducted with the control group to obtain their views on physical activity inside and outside of school. The workshops will take place once a year during each year of data collection”
Comment #20: Will you record if the class teacher changes. Will you do any observations of the teachers and how they actually implement TDM – this is likely to different within each class. What impact might this have? E.g., a teacher than runs with them compared to a teacher that takes an umbrella out in the rain and stands there and shouts at the children to keep going but doesn't get involved.	As the class teacher will need to complete questions about the child in each year of the study, we will be recording a change in the teacher. Teachers involvement in TDM activities will be obtained through the teacher questionnaire.	
Comment #21: Will you record if the schools engaging in any other whole school approaches to PA? This might be important in case there is a shift away from TDM to something else.	The school survey has captured other whole school approaches to physical activity. Other usual physical activities that classes engage in will be captured in the teacher questionnaire.	

References

1. Department of Health and Social Care. Physical activity guidelines: UK Chief Medical Officers' Report. GOV.UK; 2019.
2. The Daily Mile Foundation. The Daily Mile [Available from: <https://thedailymile.co.uk/>].

3. Marmot M, Allen J, Boyce T, et al. Health equity in England: The Marmot Review 10 years on. London: Institute of Health Equity; 2020.
4. Chesham RA, Booth JN, Sweeney EL, et al. The Daily Mile makes primary school children more active, less sedentary and improves their fitness and body composition: a quasi-experimental pilot study. *BMC Medicine*. 2018;16(1):64.
5. Morris JL, Daly-Smith A, Archbold VS, et al. The Daily Mile (TM) initiative: Exploring physical activity and the acute effects on executive function and academic performance in primary school children. *Psychol Sport Exerc*. 2019;45.
6. Chalkley AE, Routen AC, Harris JP, et al. "I Just Like the Feeling of It, Outside Being Active": Pupils' Experiences of a School-Based Running Program, a Qualitative Study. *J Sport Exerc Psychol*. 2020 Feb 1;42(1):48–58
7. Marchant E, Todd C, Stratton G, et al. The Daily Mile: Whole-school recommendations for implementation and sustainability. A mixed-methods study. *PLoS One*. 2020;15(2):e0228149.
8. Booth JN, Chesham RA, Brooks NE, et al. A citizen science study of short physical activity breaks at school: improvements in cognition and wellbeing with self-paced activity. *BMC Med*. 2020;18(1):62.
9. Brown HE, Whittle F, Jong ST, et al. A cluster randomised controlled trial to evaluate the effectiveness and cost-effectiveness of the GoActive intervention to increase physical activity among adolescents aged 13–14 years. *BMJ Open* 2017;7:e014419.
10. Collings PJ, Wijndaele K, Corder K, et al. Levels and patterns of objectively-measured physical activity volume and intensity distribution in UK adolescents: the ROOTS study. *Int J Behav Nutr Phys Act*. 2014;11:23.
11. Lloyd J, Creanor S, Logan S, et al. Effectiveness of the Healthy Lifestyles Programme (HeLP) to prevent obesity in UK primary-school children: a cluster randomised controlled trial. *The Lancet Child & Adolescent Mental Health*. 2018; 2: 35–45.
12. Duncan MJ, Wilson S, Tallis J, et al. Validation of the Phillips et al. GENEActiv accelerometer wrist cut-points in children aged 5–8 years old. *Eur J Pediatr*. 2016 Dec;175(12):2019-2021.

Appendix 1: Supplemental Table 2 (amended): Outcomes and measuring tools included in the iMprOVE study

Outcome	Measured by	Outcome/completed by
Physical activity ¹	 • Device-based (accelerometer) 	Children's physical activity
	 • Children's Physical Activity Questionnaire (C-PAQ)^a Parent 	Parent/Carer ² to complete about their child
Mental health	 • International Physical Activity Questionnaire (IPAQ)^b 	Parent/Carer ² physical activity
	 • The Warwick and Edinburgh Mental Wellbeing Scale (WEMWBS)^c • Strengths and Difficulties Questionnaire (SDQ)^e 	Parent/Carer ² mental health Parent/Carer ² and Teacher to complete about the child
Educational attainment	 • Likert scale from 1 (below expected levels) to 5 (above expected levels)^f 	Teacher's ratings
General health and wellbeing	 • Child Health Utility 9D (CHU-9D)^d 	Parent/Carer ² to complete for their children in school years 1 and 2. For children in years 3 to 6, children will be encouraged to

	 • EuroQol Visual Analogue Scale (EuroQol VAS)^g 	complete questions by themselves with help from parent/carer.
	 • EuroQol Descriptive Scale^g 	Parent/Carer ² general health, and parent/carer to complete for their child in school years 1 and 2. Children in years 3 to 6, children will be encouraged to complete questions by themselves with help from parent/carer.
	 • Questions from the UK Census 2011^h 	Parent/Carer ² general health, and parent/carer to complete for their child in school years 1 and 2. Children in years 3 to 6, children will be encouraged to complete questions by themselves with help from parent/carer.
	 • Life satisfaction, worthwhile, happiness and anxiety questions from Measuring National Wellbeing (MNW)ⁱ 	Parent/Carer ² wellbeing
Socio-demographic	 • Questions from the UK Census 2011^h 	Parent/Carer ² own socio-demographic data, and parent/carer to complete socio-demographic data about their child if their child is in school years 1 or 2. Children in years 3 to 5 will be encouraged to complete these questions on their own with help from their parent/carer.
Anthropometry	 • Height measure • Bioimpedance scales 	Measured by study researchers.

¹TDM physical activity: when taking place and teacher involvement will be included in the teacher questionnaire.

²Any parent/carer for the child can complete the questionnaires at baseline. For consistency, we will ask the parent/carer who completed the questionnaire at baseline to complete questionnaires at each follow-up.

^aKowalski k, Crocker R and Donen R. The Physical Activity Questionnaire for Older Children (PAQ-C) and Adolescents (PAQ-Q) Manual. University of Saskatchewan.

^bCraig, C. L., et al. (2003). "International physical activity questionnaire: 12-country reliability and validity." *Med Sci Sports Exerc* 35: 1381-95.

^cTenant R, Hiller L, Fishwick R et al. The Warwick-Edinburgh Mental Well-being Scale (WEMWBS): development and UK validation. *Health Qual Life Outcomes*. 2007;5. DOI: 10.1186/1477-7525-5-63.

^dStevens KJ. Working with Children to Develop Dimensions for a Preference-Based, Generic, Paediatric, Health-Related Quality-of-Life Measure. *Qual Health Res.* 2010;20(3):340-351. DOI: 10.1177/1049732309358328.

^eGoodman R. The Strengths and Difficulties Questionnaire: a research note. *Journal of Child Psychology and Psychiatry.* 1997;38(5):581-586. DOI: 10.1111/j.1469-7610.1997.tb01545.x.

^fBrehehy K, Passmore S, Adab P, et al. Effectiveness and cost-effectiveness of The Daily Mile on childhood weight outcomes and wellbeing: a cluster randomised controlled trial. *International Journal of Obesity.* 2020;44(4):812-822. DOI: 10.1038/s41366-019-0511-0.

^gEuroQol Research Foundation. EQ-5D-5L User Guide 2019.

^hUK Census United Kingdom. Office for National Statistics (ONS). 2011.

ⁱMeasuring National Wellbeing – Personal Wellbeing. Office for National Statistics, 2019

VERSION 2 – REVIEW

REVIEWER	Hanna Nalecz Institute of Mather and Child, Poland
REVIEW RETURNED	16-Mar-2021

GENERAL COMMENTS	Review bmjopen-2020-045879.R1 Impact of The Daily Mile™ on children’s physical and mental health and educational attainment in primary schools; iMprOVE cohort study protocol Dear Authors, After the amendments and supplementation indicated by the Reviewers in the first wave were introduced, the manuscript became more transparent. The care for explaining all the details is also noticeable now. Authors’ responses to reviewers’ comments are clear and accurate and were implemented according to the provided guidelines. Therefore, I recommend the protocol to be published in current form. Kind regards, HNalecz
---

REVIEWER	Dr Jade Lynne Morris King's College London
REVIEW RETURNED	07-Apr-2021

GENERAL COMMENTS	Many thanks to the authors for making the relevant amendments to the article. I had two very minor existing points. Otherthan that, I think the changes are excellent and look forward to reading more about the progression of
--